# Unraveling the Role of RNA-Binding Proteins, with a Focus on RPS5, in the Malignant Progression of Hepatocellular Carcinoma

**DOI:** 10.3390/ijms25020773

**Published:** 2024-01-07

**Authors:** Chongyang Zhou, Qiumin Wu, Haibei Zhao, Ruixi Xie, Xin He, Huiying Gu

**Affiliations:** The Key Laboratory of Molecular Biology of Infectious Diseases Designated by the Chinese Ministry of Education, Chongqing Medical University, Chongqing 400016, China; z15215155415@163.com (C.Z.); 15270765320@163.com (Q.W.); z19963668883@163.com (H.Z.); xruixi66@163.com (R.X.); 18875273680@163.com (X.H.)

**Keywords:** hepatocellular carcinoma, RNA-binding protein, RPS5, comprehensive analysis

## Abstract

Hepatocellular carcinoma (HCC) represents a major global health concern, demanding a thorough understanding of its molecular mechanisms for effective therapeutic strategies. RNA-binding proteins (RBPs) play critical roles in post-transcriptional gene regulation, with their dysregulation increasingly recognized as a hallmark of various cancers. However, the specific contributions of RBPs to HCC pathogenesis and prevention remain incompletely understood. In this study, we systematically conducted an examination of the expression profiles and clinical relevance of RBPs in 556 clinical samples from well-established cohorts. Through comprehensive analyses, a subset of RBPs exhibiting significant overexpression in HCC was identified, establishing a noteworthy correlation between their aberrant expression and HCC progression. Furthermore, 40S ribosomal protein S5 (RPS5), a ribosomal protein, emerged as a potential key contributor in HCC progression. Rigorous analyses established a correlation between elevated RPS5 expression and advanced clinicopathological features, suggesting its potential as a prognostic biomarker. Experiments further confirmed the impact of RPS5 on pivotal cellular processes implicated in cancer progression, including cell proliferation and metastasis. Further mechanistic studies unveiled the potential of RPS5 to activate the cell cycle by binding to key molecules involved in the pathway, thereby promoting the malignant progression of HCC. Additionally, our analysis of the etiology behind RPS5 overexpression in HCC posited it as an outcome of transcriptional regulation by the transcription factors Nuclear Respiratory Factor 1 (NRF1) and MYC-associated zinc finger protein (MAZ). In conclusion, our study contributes to the growing evidence elucidating the intricate involvement of RBPs, exemplified by RPS5, in the malignant progression of HCC. The integration of genomic, transcriptomic, and functional analyses provides a comprehensive understanding of the regulatory mechanisms associated with RPS5 in HCC. This comprehensive analysis not only advances our knowledge of the molecular drivers behind HCC but also highlights the potential therapeutic relevance of targeting RBPs and their regulatory network for the development of more effective treatment strategies.

## 1. Introduction

Primary liver cancer ranks sixth in incidence and third in total mortality worldwide with dismal clinical outcomes, accounting for over 830,000 deaths in 2020 [1]. Hepatocellular carcinoma (HCC), the predominant histological subtype of primary liver cancer, is a common aggressive malignancy with complex pathogenesis and unclear prognostic biomarkers [2]. Despite significant advances in early diagnosis and innovative therapeutic strategies, HCC patients remained at high risk of postoperative recurrence and unsatisfactory survival outcomes [2,3]. Consequently, the limited efficacy of targeted therapy and inadequate clinical management indicate the necessity for further investigations into the underlying molecular mechanisms of HCC pathogenesis, with the ultimate goal of enhancing treatment outcomes.

RNA-binding proteins (RBPs) are dynamic and versatile proteins engaged in multiple layers of post-transcriptional gene expression, playing a crucial role in maintaining cellular homeostasis [4,5,6]. These RBPs possess the ability to bind RNA through one or multiple globular RNA-binding domains (RBDs), and determine the fate and function of the bound RNAs, mainly by regulating essential RNA metabolism processes, including RNA splicing, polyadenylation, stability, localization, translation, and degradation [7,8]. Published reports have highlighted the significant involvement of aberrant RBP expression in cancer development, particularly in HCC [9,10,11]. It is now evident that the imbalance between RBPs and target genes could impact various hallmarks of cancer, including tumor proliferation, metastasis, resistance to cell death, metabolic disorders, immune evasion, genomic instability, and ultimately tumor progression [9,11,12]. Thus, deciphering the intricate network of interactions between RBPs and their cancer-related RNA targets will provide a better understanding of tumor biology and potentially unveil new targets for cancer therapy. However, the critical roles of RBPs in the pathogenesis and prevention of HCC have not been fully understood.

Human 40S ribosomal protein S5 (RPS5), a component of the 40S ribosomal subunit, is mainly involved in ribosomal maturation and translation initiation by interacting with initiation factors [13]. RPS5 also has various extra-ribosomal functions, including DNA repair, apoptosis modulation, cell signaling, transcriptional regulation, and cellular transformation [14,15,16]. In addition, both the dysregulation and intrinsic dysfunctions of ribosomes result in an elevated incidence of tumors [9]. However, there is still a lack of research on the potential role of RPS5 in the cellular functions of HCC.

In this study, we aimed to identify RBPs that may play important roles in HCC. In short, we identified a set of candidate RBPs associated with HCC progression by performing bioinformatics analysis using well-established HCC datasets, and RPS5 was selected for further experimental validation due to its abundant expression and uncharacterized role in HCC. We demonstrated that RPS5 is frequently up-regulated in HCC and acts as a novel oncogenic RBP in HCC progression. Mechanistically, the overexpression of RPS5 may exert its oncogenic effect by regulating the cell cycle. Taken together, the therapeutic targeting of RPS5 may offer options for HCC intervention.

## 2. Results

### 2.1. Aberrant Expression of RBPs in HCC

In order to comprehensively elucidate the contributions of aberrant RBPs expression to HCC carcinogenesis, we systematically analyzed the expression profile and clinical relevance of all known RBP-encoding genes, utilizing data from 556 clinical samples obtained from HCC patients across well-established cohorts, including the TCGA and GEO datasets (GSE144269, GSE50579, GSE45267). Initially, we identified 1281 co-differentially expressed genes (co-DEGs) between HCC and para-tumor tissues, providing a comprehensive view of the transcriptional landscape of HCC (Figure 1A and Appendix A). Within this landscape, 288 differentially expressed RBPs (DERBPs) were identified by intersecting the co-DEGs with a comprehensive collection of 2961 known RBPs (Figure 1B, Appendix A). Remarkably, the subset of DERBPs accounted for approximately one fifth of the total co-DEGs, underscoring the pivotal role of RBPs in driving HCC progression. To elucidate the functional significance of the 288 DERBPs, KEGG and GO functional enrichment analyses were performed, revealing significant enrichment in critical pathways such as cell cycle, DNA replication, metabolism, and apoptosis (Figure 1C,D). Recognizing the fundamental role of RBPs as integral components of RNA-based regulatory mechanisms, we focused on the subset of 123 genes enriched with RNA-binding functions according to the GO Molecular Function (GO-MF) category. The expression heatmap analysis revealed that only 5 out of the 123 DERBPs exhibited down-regulation, and highlighted consistent up-regulation across the four HCC datasets for the majority of DERBPs (Figure 1E). This observation strongly suggests that the majority of DERBPs likely function as cancer-promoting factors in the context of HCC advancement.

To identify pivotal RBPs driving HCC progression, we conducted a deeper analysis of the 123 DERBPs using WGCNA, a powerful tool that groups genes with similar expression patterns into distinct color blocks [17]. Firstly, three regulatory modules were identified through clustering deprogram analysis (Appendix A). Subsequently, the relationship between these modules and various HCC risk factors, including tumor stage, histological grade, peritumoral inflammation, tumor vascular infiltration status, and alpha-fetoprotein levels were explored. The analysis revealed a robust correlation between the blue module and severe HCC (Figure 1F). Furthermore, a heatmap analysis demonstrated the significant up-regulation of the 37 genes within the blue module across all four datasets, emphasizing the potential significance of these DERBPs in driving HCC progression (Figure 1G).

### 2.2. The Up-Regulation of RPS5 in Tumor Samples Indicates HCC Malignancy

To further elucidate the intricate interactions of DERBPs within the blue module and identify key contributors in HCC, a protein–protein interaction (PPI) network was constructed using the STRING database and visualized through Cytoscape software (v3.10.0). Within the network, 36 proteins in the blue module exhibited interactions, with NHP2 (Nucleolar protein family A member 2), RPS5 (40S ribosomal protein S5), and SNRPD2 (Small nuclear ribonucleoprotein Sm D2) prominently featured (Figure 2A). Subsequently, the CytoHubba plugin, integrating ten algorithms to score genes within the PPI network, was applied to identify hub genes. The intersection of the top 10 genes from each algorithm revealed four common genes: RPS5, NHP2, RPL8 (60S ribosomal protein L8), and RPL27 (60S ribosomal protein L27) (Figure 2B). Remarkably, these genes were closely associated with ribosomal function, indicating their potential significance in driving HCC progression. In order to assess the correlation of these four RBPs with HCC malignancy, including tumor stage, pathological grade, survival rate, and metastasis, comprehensive analyses were conducted using public databases, and the results demonstrated a positive correlation between the levels of these four RBPs and HCC stage and grade (Figure 2C,D). Prognostic outcomes from TCGA-LIHC data in the KM-plot database indicated a substantial negative correlation between the high expression of these four genes and overall survival (OS) and recurrence-free survival (RFS) in HCC patients (Figure 2E,F). Furthermore, the analysis of the GSE40367 dataset, encompassing metastatic and non-metastatic HCC samples, revealed significantly elevated levels of RPS5 and NHP2 in metastatic HCC tissues (Figure 2G,H). A radar map, utilizing -log10 (*p*-value) values of these four genes as coordinates, emphasized that RPS5 occupied the largest area, indicating its potential as a distinctive indicator of HCC malignancy (Figure 2I). To further validate these findings, additional HCC datasets were employed and confirmed that RPS5 exhibited aberrantly elevated expression at the DNA, mRNA, and protein levels in HCC patient tissues compared to normal liver tissue (Appendix A). A consistent elevation in the mRNA level of RPS5 was also observed in the majority of HCC cell lines compared to normal liver cells (Appendix A). Therefore, our comprehensive analysis indicates that RPS5 displays abnormal overexpression across multiple HCC datasets, demonstrating a remarkably high correlation with factors such as tumor stage, histological grade, prognosis, and metastasis. These findings suggest a pivotal role for RPS5 as an oncogenic factor in the progression of HCC.

### 2.3. NRF1 and MAZ Are Potential Transcription Factors of RPS5 in HCC

To decipher the underlying reasons for the conspicuously elevated expression of RPS5 in HCC, our investigation initially centered on the analysis of gene mutations. The examination of data from the GSCA database and the cBioPortal database suggests that mutations are unlikely to be the causative factor for the abnormal overexpression of RPS5 in HCC (Appendix A). Transcription factors (TFs) play a pivotal role in regulating gene expression, determining cell function and shaping responses to environmental challenges [18,19]. Considering this, we sought to explore the potential involvement of transcription factors in the anomalous expression of RPS5 in HCC. Initially, we employed the JASPAR plugin within the UCSC dataset to predict TFs in the RPS5 promoter region, revealing a list of 12 TFs (Figure 3A). To narrow down our focus, we intersected these TFs with those relevant to the human liver in the Cistrome DB database, identifying three significant transcription factors: NRF1 (Nuclear Respiratory Factor 1), MAZ (MYC-associated zinc finger protein), and TCF12 (transcription factor 12) (Figure 3B). Subsequently, utilizing the Cistrome DB database, we assessed the likelihood of these three transcription factors targeting RPS5, yielding predicted scores of NRF1 (2.694 points), MAZ (1.799 points), and TCF12 (0.036 points), respectively (Figure 3C). Consequently, NRF1 and MAZ were postulated as potential transcription factors governing RPS5. NRF1, a nuclear transcription factor, exhibits diverse functions in regulating oxidative stress, endoplasmic reticulum stress, inflammation, and immune modulation [20,21]. Similarly, MAZ, a C2H2-type zinc finger protein, orchestrates the transcription of various genes, including matrix metalloproteinases and serum amyloid A, playing crucial roles in metabolic diseases, cancers, and other pathological conditions [22,23]. Furthermore, motif sequences of NRF1 and MAZ binding to RPS5 were predicted through the JASPAR database (Figure 3D). Finally, the expression levels of NRF1 and MAZ, along with their correlation with RPS5 were analyzed in selected HCC datasets, and the observations revealed that both NRF1 and MAZ were conspicuously overexpressed in HCC compared to normal liver tissue, and their expression levels exhibited a notable positive correlation with RPS5 expression (Figure 3E and Appendix A). These findings suggest that the heightened expression of transcription factors NRF1 and MAZ in HCC could be a contributing factor to the abnormal overexpression of RPS5.

### 2.4. RPS5 Promotes the Progression of Liver Cancer by Regulating Cell Cycle

To elucidate the mechanisms underlying the promotion of malignant progression in HCC by RPS5, we initially integrated four HCC datasets employed in this study, effectively eliminating batch effects (Appendix A). Subsequently, we sorted the RNAseq data of 556 HCC patients based on RPS5 expression levels, categorizing them into the top 1/4 with high RPS5 expression (139 cases) and the bottom 1/4 with low RPS5 expression (139 cases) for differential gene analysis. This analysis unveiled 2654 DEGs co-expressed with RPS5 (*p* value < 0.05) (Figure 4A and Appendix A). Furthermore, by employing the catRAPID database to predict RNAs binding to RPS5, we obtained 5794 RNAs with a ranking score greater than 0.6 (Appendix A). The intersection of the 2654 DEGs co-expressed with RPS5 and the 5794 RNAs predicted to bind with RPS5 resulted in 648 genes (Figure 4B and Appendix A). To narrow down the focus on molecules pivotal in RPS5-promoted HCC progression, we conducted a disease enrichment analysis using the DISGENET database on these 648 genes, revealing that 64 genes significantly enriched in liver carcinoma (Figure 4C). Furthermore, KEGG and GO functional enrichment analyses were performed on these 64 genes, indicating a significant enrichment in the cell cycle pathway (Figure 4D,E). The GSEA enrichment results similarly showed a significant activation of the cell cycle pathway in HCC patients with a high RPS5 expression (Figure 4F). Moreover, by utilizing the Panther database to categorize the 31 genes enriched in the cell cycle, it was found that these proteins are primarily involved in enzymes, protein-binding proteins, metabolic pathway proteins, cytoskeletal proteins, and transcription factors (Figure 4G,H). This suggests that RPS5 may activate the cell cycle through complex molecular mechanisms, thereby regulating the malignant progression of HCC.

### 2.5. Experimental Confirmation of the Frequent Up-Regulation of RPS5 in HCC

The preliminary clinicopathological correlation analysis indicated that RPS5 exhibited a higher expression in HCC tissues compared to noncancerous hepatic tissues. Subsequently, validation was performed using clinical HCC liver tissue samples and liver cancer cells. Initially, RPS5 mRNA levels were assessed using qRT-PCR in HCC and non-tumor tissues, revealing a significant up-regulation in 57.5% (23/40) of HCC samples (Figure 5A,B). A further classification of HCC samples also indicated a higher RPS5 expression in samples with metastasis (EHMH) than in samples without metastasis (MFH) (Figure 5C). Correspondingly, Western blot experiments and IHC staining corroborated these results, confirmed elevated protein levels of RPS5 in HCC tissues compared to para-tumor tissues (Figure 5D,E). To validate the clinical significance of these observations, we investigated the expression profiles of RPS5 in a panel of HCC cell lines, observing a lower expression in primary human hepatocytes (PHH) and a higher expression in HCC cell lines (Figure 5F–H). Collectively, these data underscore the clinical association of RPS5 dysregulation with the biology and severity of HCC.

### 2.6. RPS5 Promotes Hepatic Tumorigenesis In Vitro and In Vivo

Subsequently, we stably silenced RPS5 in MHCC97H and HLE cells using shRNA-encoding lentiviruses, and the knockdown efficiency was confirmed using qRT-PCR and Western blotting (Figure 6A). The reduction in RPS5 expression led to a notable decrease in cell proliferation, as evidenced by the CCK-8 assay (Figure 6B). This observation was further substantiated by the colony formation assay, revealing a significant repression of the colony formation ability in MHCC97H and HLE cells following RPS5 knockdown (Figure 6C). To elucidate the mechanism underlying the anti-proliferative effects mediated by RPS5 down-expression in HCC, we evaluated cell cycle distribution using flow cytometry, revealing a G1 arrest following RPS5 knockdown in HCC cells (Figure 6D). Given previous analyses indicating elevated RPS5 expression in metastatic HCC tissues, we conducted both wound-healing and transwell assays to investigate the impact of RPS5 on the metastasis of HCC. The results demonstrated that RPS5 knockdown inhibited the wound-healing ability and migration of MHCC97H and HLE cells (Figure 6E,F). Recognizing the crucial role of cellular filaments, constituting the cellular framework, in augmenting cell motility and contributing to tumor metastasis, we further utilized the specific marker phalloidin to stain and visualize intracellular filaments, revealing an uneven distribution and a significant reduction in filament quantity following RPS5 knockdown (Figure 6G).

Subsequently, a gain-of-function study was conducted to probe the effects of RPS5 overexpression on HCC progression. The results confirmed that RPS5 overexpression in Huh7 and PLC/RPF/5 cells promoted cell proliferation and colony formation while inhibiting G1 arrest (Figure 7A–D). The wound-healing assay, transwell experiments, and phalloidin staining additionally indicated a significant enhancement in cell motility upon RPS5 overexpression. (Figure 7E–G).

To further assess the impact of RPS5 on HCC tumorigenesis in vivo, we injected MHCC97H cells with stable RPS5 knockdown or control cells into the livers of nude mice, facilitating an examination of tumor growth in situ and lung metastasis. Remarkably, in comparison to mice injected with control cells, those injected with RPS5-silenced HCC cells exhibited dramatically inhibited tumorigenesis and lung metastasis (Figure 8). In summary, these results demonstrated that RPS5 promotes HCC tumorigenesis both in vitro and in vivo.

## 3. Discussion

The malignant progression of tumors represents a dynamic process governed by complex molecular interactions, with RBPs emerging as critical orchestrators in post-transcriptional gene regulation [9,11,24]. In this study, we identified a subset of RBPs significantly overexpressed in HCC, establishing a noteworthy correlation between their aberrant expression and the progression of HCC. The dysregulation of RBPs has been increasingly recognized as a hallmark of various cancers, and our findings contribute to the growing understanding of their involvement in HCC pathogenesis.

Our investigation specifically identified RPS5, a ribosomal protein, as a potential contributor in HCC progression. Through comprehensive analyses, we established a robust association between elevated RPS5 expression and advanced clinicopathological features, including tumor stage, histological grade, and prognosis, indicative of its potential as a prognostic biomarker. In vivo and in vitro experiments further confirmed the impact of RPS5 on pivotal cellular processes implicated in cancer progression, including cell proliferation and metastasis, suggesting its involvement in the unrestrained growth characteristic of malignancy. Further mechanistic studies unveiled the potential of RPS5 to activate the cell cycle by binding to key molecules involved in the progression. Therefore, the comprehensive analysis and experimental results highlighted the multifaceted impact of RPS5 on various cellular processes in the malignant advancement of HCC, accentuating its significance as a multifaceted contributor to the complex landscape of hepatocellular carcinogenesis.

Additionally, we also analyzed the etiology contributing to the abnormally high expression of RPS5 in HCC. The comparative analysis of gene expression patterns and associated landscapes revealed potential mechanical insights. Our findings indicate that while RPS5 overexpression is notably linked to CNVs, frequent mutations in RPS5 may not significantly impact the diagnosis of HCC patients. This suggests that alternative regulatory mechanisms, such as adaptations in translational regulation, might be more prospective contributors to the abnormal overexpression of RPS5 in HCC. Consequently, we explored the role of TFs in modulating RPS5 expression, and identified NRF1 and MAZ as potential TFs. NRF1 and MAZ are established regulators of various cellular processes, including oxidative stress, inflammation, and immune modulation. The abnormal overexpression of NRF1 and MAZ in HCC, coupled with their positive correlation with RPS5 expression, implies a potential regulatory axis contributing to the pro-proliferative effects of RPS5 in HCC.

In conclusion, our study adds to the growing body of evidence highlighting the intricate involvement of RBPs, exemplified by RPS5, in the malignant progression of HCC. The integration of genomic, transcriptomic, and functional analyses provides a comprehensive understanding of the regulatory mechanisms associated with RPS5 in HCC. This research not only advances our knowledge of the molecular drivers behind HCC but also highlights the potential therapeutic relevance of targeting RBPs and its regulatory network in the pursuit of more effective treatment strategies for HCC.

## 4. Materials and Methods

### 4.1. Data Downloading and Comparative Gene Expression Analysis

The TCGA-LIHC (The Cancer Genome Atlas-Liver Hepatocellular Carcinoma) datasets, comprising 371 tumor tissues and 50 normal tissues, along with pertinent clinical information, were acquired from the Xena website (https://xenabrowser.net/datapages/, accessed on 1 September 2023). Differential gene expression analysis between the normal and tumor tissues within the TCGA-LIHC datasets was performed utilizing the limma package, accessed through the Sangerbox website (http://vip.sangerbox.com/, accessed on 1 September 2023). Furthermore, datasets GSE144269, GSE50579, GSE45267, and GSE40367, encompassing data from 185 HCC tissues with diverse etiologies and 119 normal tissues, were obtained from the GEO database (https://www.ncbi.nlm.nih.gov/gds/, accessed on 1 September 2023). Differential gene expression analysis for these four datasets was performed in the NCBI GEO portal (https://www.ncbi.nlm.nih.gov/geo/geo2r/, accessed on 1 September 2023).

### 4.2. Weighted Gene Co-Expression Network Analysis (WGCNA)

WGCNA analysis was performed utilizing the Bioinformatics and Information Center (BIC) website (https://www.bic.ac.cn/BIC/#/, accessed on 1 September 2023). This analysis aimed to group genes exhibiting coordinated expression patterns based on clinical characteristics. RNA-seq data and clinical information of HCC patients, sourced from the TCGA-LIHC datasets, constituted the dataset for this analysis.

### 4.3. Cell Culture

Human liver cancer cell lines MHCC97H and HLE were procured from the Liver Cancer Institute of Zhongshan Hospital, Fudan University (Shanghai, China). Huh-7 and PLC/RPF/5 were acquired from the Health Science Research Resource Bank. Cells were cultured in DMEM supplemented with 10% fetal bovine serum (Gibco BRL) and 1% penicillin/streptomycin. All cell lines were maintained at 37 °C in a 5% CO_2_ atmosphere. The authentication of cell lines was performed by Guangzhou Jennio Biotech Co., Ltd., Guangzhou, China.

### 4.4. Clinical Samples

A cohort comprising 40 paired human HCC liver tissues and corresponding non-tumorous liver tissues was obtained from the Second Affiliated Hospital of Army Medical University, PLA, Chongqing, China. Tissue specimens were snap-frozen in liquid nitrogen upon resection and stored at −80 °C for subsequent investigations. For immunohistochemistry staining, specimens were formalin-fixed and paraffin-embedded. Informed consent was obtained from all patients, and the study protocol adhered to the ethical guidelines outlined in the 1975 Declaration of Helsinki. Approval for the study was granted by the Ethics Committee of Army Medical University, PLA.

### 4.5. Animal Experiment

Male BALB/C nude mice (4–5 weeks old) were sourced from Beijing Huafukang Biotech (Beijing, China) and allowed a one-week adaptation period upon arrival. MHCC97H cells, stably knocked down for RPS5, at a concentration of 1.5 × 10^6^ cells were orthotopically implanted into the left hepatic lobe of nude mice using a 40 μL serum-free DMEM/Matrigel (BD Biosciences, Franklin Lakes, NJ, USA) mixture (1:1 *v*/*v*). The mice were sacrificed at 8 weeks post injection, and liver tumor volumes were calculated using the formula: V(mm^3^) = width^2^ (mm^2^) × length (mm)/2. Metastasized tumor nodules and foci in the lungs were quantified through microscopic observation. Both liver and lung tumor tissues underwent hematoxylin and eosin (HE) staining.

### 4.6. Construction of the RPS5 shRNA Lentivirus

To establish an RPS5 knockdown cell line and its non-target control, lentivirus-mediated gene editing was conducted. Small interfering RNA sequences targeting the human RPS5 gene (5′-gcAGGATTACATTGCAGTGAA-3′) were designed and cloned into the plasmid pGCL-GFP by Shanghai GeneChem Co., Ltd., Shanghai, China. A similar process was followed for constructing the non-targeting control (5′-TTCTCCGAACGTGTCACGT-3′). These modified plasmids were co-transfected into HEK293T cells with lentiviral packaging plasmids to generate RPS5 shRNA-expressing lentivirus or a control shRNA-expressing lentivirus. After the infection of the MHCC97H and HLE cells, puromycin selection (2 μg/mL) was applied for 2 weeks. Single-cell clones were isolated, subjected to TA cloning with subsequent sequencing to confirm knockout efficiency, and further assessed using Western blotting.

### 4.7. Statistical Analysis and Chart Plotting

Various visualizations, including bar graphs, scatter plots, Venn diagrams, upset plots, Venn networks, heatmaps, volcano plots, radar charts, etc., were analyzed and created using GraphPad Prism9, Chiplot, Sangerbox, Oebiotech, and Evenn. A significance level of *p* < 0.05 was applied. Unless otherwise stated, data are presented as mean ± SEM, and the Mann–Whitney U test was utilized for data analysis between two groups, while the paired t-test was applied to analyze the expression levels in HCC tissues and corresponding adjacent noncancerous tissues.

For other materials and methods, please refer to the Appendix A.

## Figures and Tables

**Figure 1 ijms-25-00773-f001:**
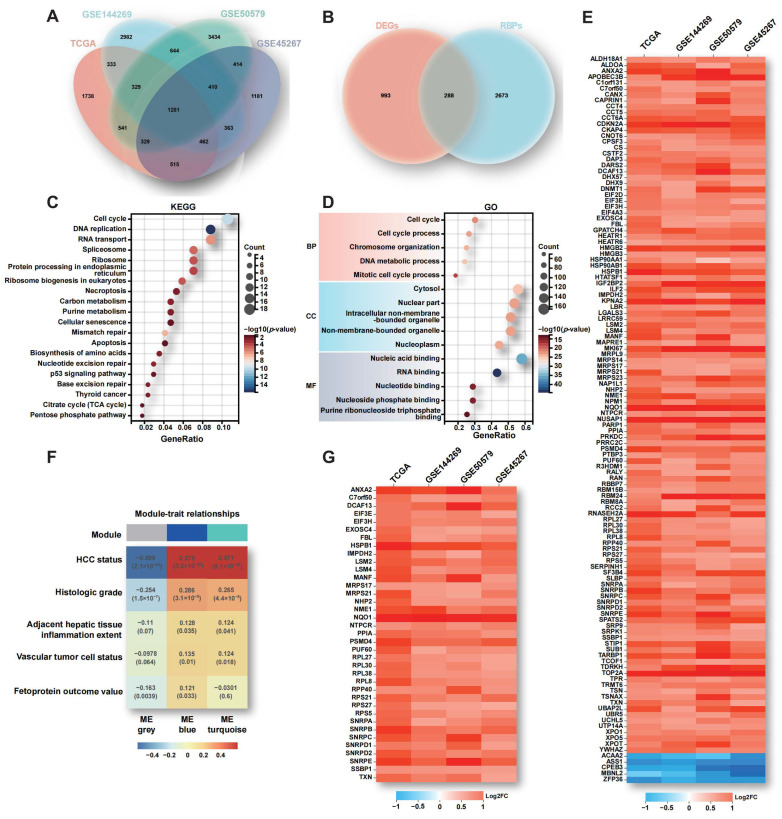
Alterations of RBPs in HCC. (**A**,**B**) The Venn diagram shows a comparison of 1281 co-DEGs and 288 DERBPs between HCC and para-tumor tissues across four HCC cohorts. (**C**,**D**) KEGG and GO functional enrichment analyses were performed on the 288 DERBPs. (**E**) A heatmap displays the subset of 123 DERBPs that are enriched with RNA-binding functions according to the GO-MF category. (**F**) Clustering dendrogram analysis using WGCNA was conducted on the 123 DERBPs. (**G**) Another heatmap presents the subset of 37 DERBPs within the blue module across all four datasets.

**Figure 2 ijms-25-00773-f002:**
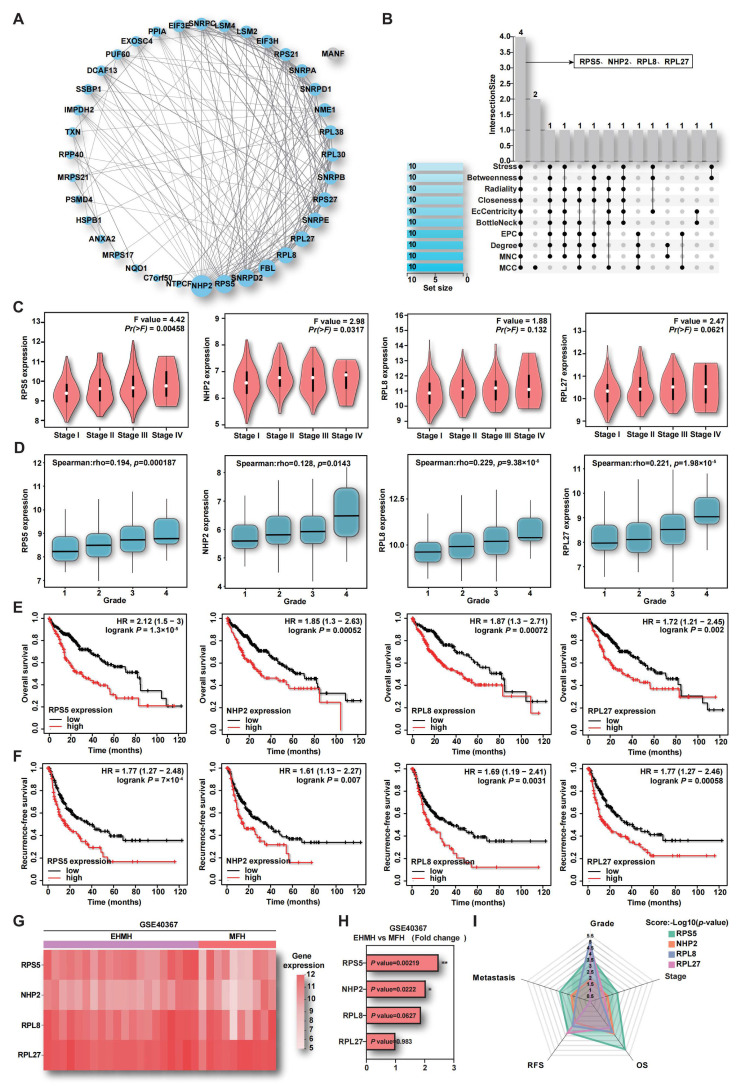
The relationship between key DERBPs and clinical data of HCC. (**A**) The PPI network shows the subset of 37 DERBPs within the blue module. (**B**) A Venn diagram illustrates the hub genes in the PPI network, identified using ten algorithms of the CytoHubba plugin. (**C**,**D**) The correlation between four DERBPs and HCC stage and grade is demonstrated. (**E**,**F**) The correlation between four DERBPs and OS and RFS in HCC patients is analyzed. (**G**,**H**) A heatmap and fold change expression analysis display the four DERBPs in the GSE40367 dataset. (**I**) A radar map is generated to highlight the significance (−log10 (*p*-value) values) of these four DERBPs in determining HCC malignancy. * *p* < 0.05, ** *p* < 0.01.

**Figure 3 ijms-25-00773-f003:**
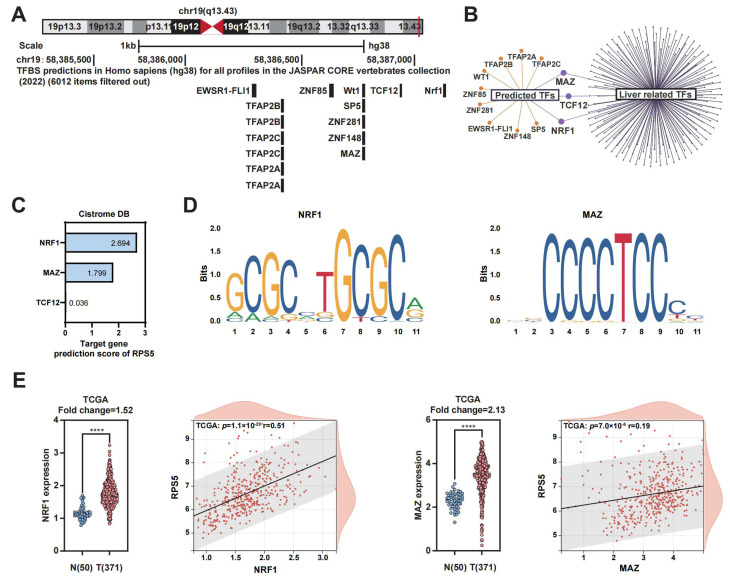
NRF1 and MAZ are potential transcription factors of RPS5 in HCC. (**A**) The JASPAR plugin within the UCSC dataset was utilized to predict transcription factors in the RPS5 promoter region. (**B**) Intersection analysis was performed to identify 3 significant transcription factors out of 12 candidates, with a focus on those relevant to the human liver in the Cistrome DB database. (**C**) The predicted scores of NRF1, MAZ, and TCF12 targeting RPS5 were assessed. (**D**) The JASPAR database was used to predict motif sequences of NRF1 and MAZ binding to RPS5. (**E**) The expression levels of NRF1 and MAZ, as well as their correlation with RPS5, were analyzed in the TCGA-LIHC datasets. The abscissa represents the expression distribution of the first gene, and the ordinate represents the expression dis-tribution of the second gene, the straight line signifies the correlation analysis conducted between the two genes. **** *p* < 0.0001.

**Figure 4 ijms-25-00773-f004:**
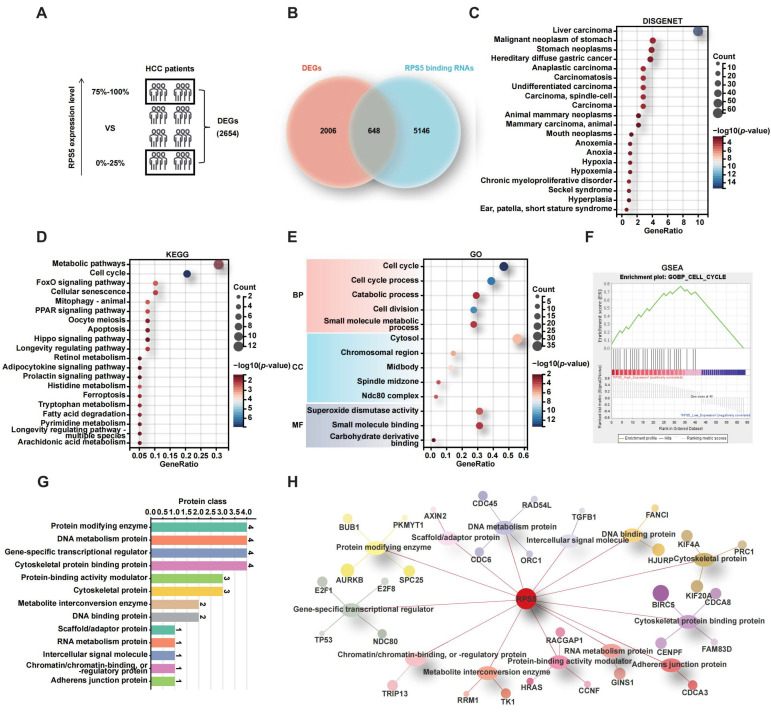
RPS5 promotes liver cancer progression through the regulation of the cell cycle. (**A**) The pattern diagram depicts the co-expression of 2654 DEGs with RPS5. (**B**) The Venn diagram highlights the intersection of 648 genes, which represent the overlap between the 2654 DEGs co-expressed with RPS5 and the 5794 RNAs predicted to bind with RPS5. (**C**) Disease enrichment analysis using the DISGENET database was conducted on the 648 genes. (**D**–**F**) KEGG pathways, GO terms, and GSEA enrichment analyses were performed to elucidate the involvement of the 64 genes enriched in liver carcinoma. (**G**,**H**) The Panther database was utilized to classify the 31 genes enriched in the cell cycle.

**Figure 5 ijms-25-00773-f005:**
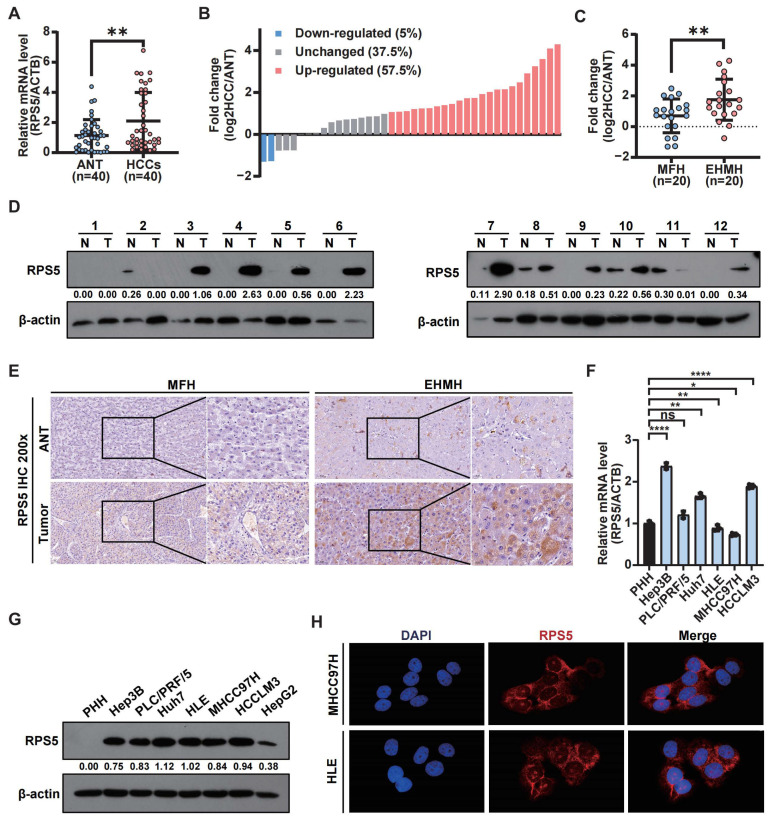
RPS5 exhibits frequent up-regulation in human HCC. (**A**,**B**) The mRNA levels of RPS5 in 40 HCC tissues and paired non-tumor tissues. (**C**) The mRNA levels of RPS5 in MFH and EHMH. (**D**) Protein levels of RPS5 in 12 HCC tissues and paired non-tumor tissues. (**E**) Immunohistochemical staining of the intensity of RPS5 expression in non-tumor and HCC tissues (magnification, 200×). (**F**,**G**) qRT-PCR and Western blot analyses of the expression of RPS5 in HCC cell lines and primary human hepatocytes (PHH). (**H**) Immunofluorescence staining of the intensity of RPS5 in MHCC97H and HLE cells (magnification, 630×). Statistical significance is denoted as “ns” indicate no significant difference, * *p* < 0.05, ** *p* < 0.01, and **** *p* < 0.0001.

**Figure 6 ijms-25-00773-f006:**
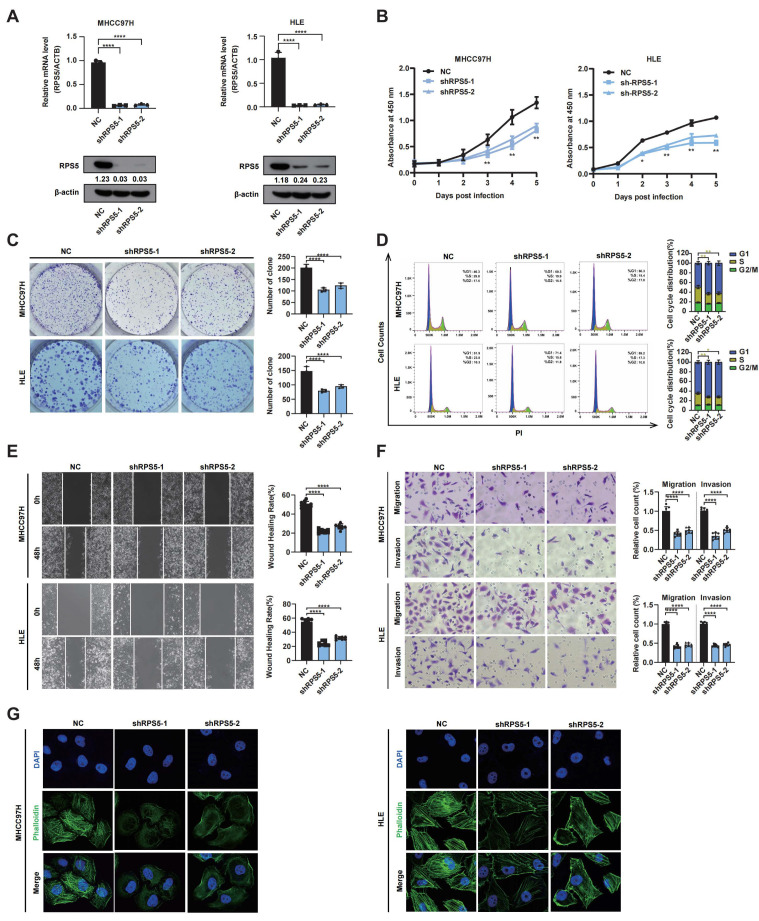
RPS5 knockdown demonstrates a suppressive effect on hepatocarcinogenesis in vitro. (**A**) qRT-PCR and Western blot analysis of the mRNA and protein levels of RPS5 in RPS5-knockdowned MHCC97H and HLE cells, respectively. (**B**) Cell proliferation, (**C**) colony formation, (**D**) cell cycle, (**E**) wound-healing, (**F**) transwell migration assays (magnification, 400×), and (**G**) phalloidin staining were performed in RPS5-knockdowned MHCC97H and HLE cells (magnification, 630×). Statistical significance is indicated as * *p* < 0.05, ** *p* < 0.01, and **** *p* < 0.0001.

**Figure 7 ijms-25-00773-f007:**
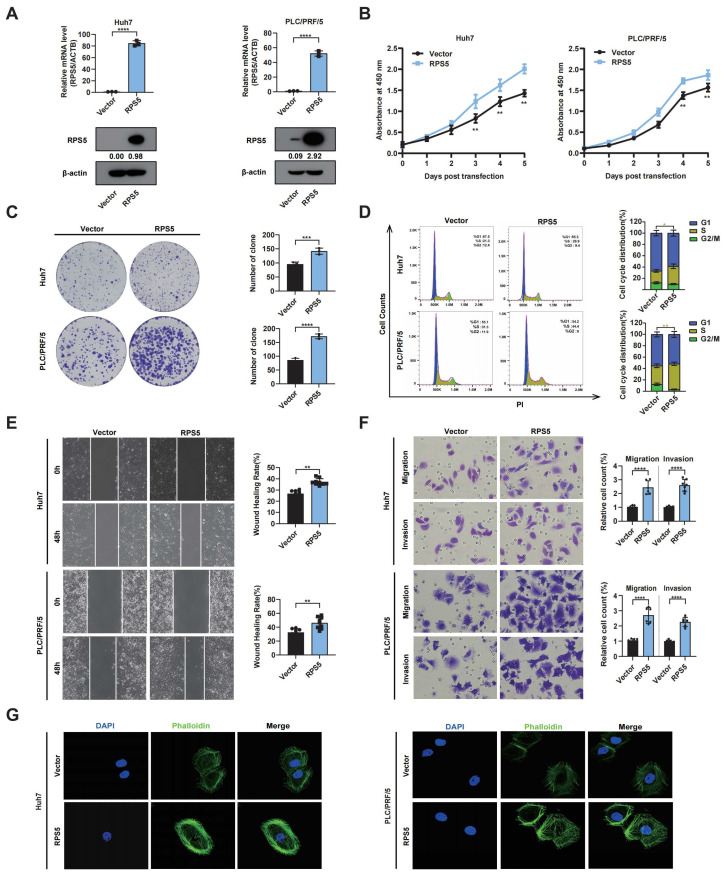
RPS5 overexpression promotes hepatocarcinogenesis in vitro. (**A**) qRT-PCR and Western blot analysis of the mRNA and protein levels of RPS5 in RPS5-overexpressed Huh7 and PLC/RPF/5 cells, respectively. (**B**) Cell proliferation, (**C**) colony formation, (**D**) cell cycle, (**E**) wound-healing, (**F**) transwell migration assays (magnification, 400×), and (**G**) phalloidin staining were performed in RPS5-overexpressed Huh7 and PLC/RPF/5 cells (magnification, 630×). Statistical significance is indicated as * *p* < 0.05, ** *p* < 0.01, *** *p* < 0.001, and **** *p* < 0.0001.

**Figure 8 ijms-25-00773-f008:**
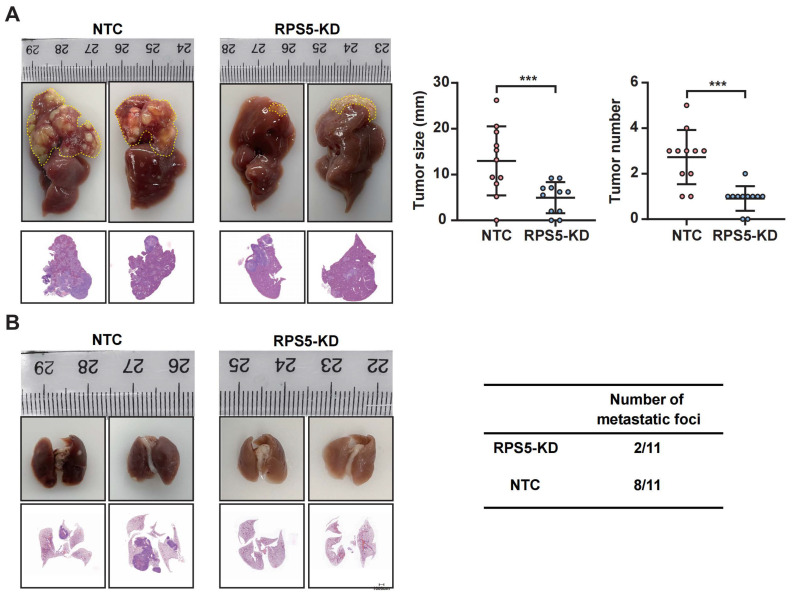
RPS5 knockdown shows inhibitory effects on HCC cell proliferation and metastasis in vivo. (**A**,**B**) Orthotopic mouse models were constructed using RPS5-knockdown MHCC97H cells and control cells (each group, n = 11). The effect of RPS5-knockdown on tumor size, tumor number, and lung metastasis was evaluated. Representative data are presented from at least three independent experiments. Data are shown as mean ± SD. Statistical significance is denoted as *** *p* < 0.001.

## Data Availability

The original contributions presented in this study are included in the article. Further inquiries can be directed to the corresponding author.

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
