# Peer review of "Unraveling the Role of RNA-Binding Proteins, with a Focus on RPS5, in the Malignant Progression of Hepatocellular Carcinoma"

_ijms, 2024, doi:10.3390/ijms25020773_

Round 1

Reviewer 1 Report

Comments and Suggestions for Authors

The purpose of this study was to undertake a thorough analysis of RBP expression patterns and clinical significance in 559 clinical samples from well-established cohorts. A group of RBPs displaying considerable overexpression in HCC was found by extensive investigations, indicating a notable link between their abnormal expression and HCC development. Actually, the current proposal is interesting and well-written. Therefore, I recommend that the current study be published after major revisions as follows:

1- The authors mentioned that ‘’ Consequently, we explored 348 the role of TFs in modulating RPS5 expression, and identified NRF1 and MAZ as potential 349 TFs. NRF1 and MAZ are established regulators of various cellular processes, including 350 oxidative stress, inflammation, and immune modulation’’ Did the authors investigate any role for immune modulation?

2- Please provide densitometric analysis for all western blots, such as figures 5D and G.

3- Please mention the 5H magnification power.

4- How did the authors choose the cell lines MHCC97H and HCCLM3?

5- Please provide the entire name of each abbreviation in the abstract section.

6- Please modify the references to meet the journal's criteria.

Reviewer 2 Report

Comments and Suggestions for Authors

I think that the manuscript submitted for possible publication, can be accepted after minor revision.

It is an interesting look to relevant role of biomarkers in the development of HCC. In any case, I think that is important to specify, if available (differently as a limitation), the possible role or confounding factor of etiology of liver cancer.

Reviewer 3 Report

Comments and Suggestions for Authors

Our esteemed authors,

As it is known, the frequency of hepatocellular carcinoma (HCC) has increased recently due to reasons such as viral hepatitis, alcohol and NASH, and it has become a serious health problem.

Studies are trying to understand the molecular mechanism of HCCs, determine the prognosis and understand their behavior such as metastasis, and thus try to treat and control them.

In this study conducted by esteemed authors on a large number of samples, they examined the behavior of a specific ribonucleid-binding protein in HCC on the course of HCC and its ability to metastasize, and found a correlation between them. This will allow us to make great progress in the monitoring and treatment of HCC in the future.

I believe that the publication of this article, which has a very high scientific content, in a good journal will contribute greatly to the education of medical students, assistants, specialists and even professors, and to patient follow-up and treatment.

I would like to thank our esteemed authors for this beautiful work and wish them success in their work.

Round 2

Reviewer 1 Report

Comments and Suggestions for Authors

The authors have successfully addressed all comments